# Understanding interactions between risk factors, and assessing the utility of the additive and multiplicative models through simulations

**Lina-Marcela Diaz-Gallo**[1], **Boel Brynedal**[2], **Helga Westerlind**[3], **Rickard Sandberg**[4], **Daniel Ramsköld**[4] *

**1** Division of Rheumatology, Department of Medicine Solna, Center for Molecular Medicine, Karolinska Institutet, Karolinska University Hospital, Stockholm, Sweden, **2** Institute of Environmental Medicine, Karolinska Institutet, Stockholm, Sweden, **3** Clinical Epidemiology Division, Department of Medicine Solna, Karolinska Institutet, Stockholm, Sweden, **4** Department of Cell and Molecular Biology, Karolinska Institutet, Stockholm, Sweden

* daniel.ramskold@ki.se

**Data Availability Statement:** Pre-existing genetic data from the EIRA cohort was used. Due to ethical considerations, data from EIRA cannot be publicly shared. Please contact the EIRA's principal

## Abstract

Understanding the genetic background of complex diseases requires the expansion of studies beyond univariate associations. Therefore, it is important to use interaction assessments of risk factors in order to discover whether, and how genetic risk variants act together on disease development. The principle of interaction analysis is to explore the magnitude of the combined effect of risk factors on disease causation. In this study, we use simulations to investigate different scenarios of causation to show how the magnitude of the effect of two risk factors interact. We mainly focus on the two most commonly used interaction models, the additive and multiplicative risk scales, since there is often confusion regarding their use and interpretation. Our results show that the combined effect is multiplicative when two risk factors are involved in the same chain of events, an interaction called synergism. Synergism is often described as a deviation from additivity, which is a broader term. Our results also confirm that it is often relevant to estimate additive effect relationships, because they correspond to independent risk factors at low disease prevalence. Importantly, we evaluate the threshold of more than two required risk factors for disease causation, called the multifactorial threshold model. We found a simple mathematical relationship (square root) between the threshold and an additive-to-multiplicative linear effect scale (AMLES), where 0 corresponds to an additive effect and 1 to a multiplicative. We propose AMLES as a metric that could be used to test different effects relationships at the same time, given that it can simultaneously reveal additive, multiplicative and intermediate risk effects relationships. Finally, the utility of our simulation study was demonstrated using real data by analyzing and interpreting gene-gene interaction odds ratios from a rheumatoid arthritis case-control cohort.

investigators for data requests for applicable studies. For further information go to: https://www.eirasweden.se/Kontakt_EIRA.htm.

**Funding:** L.M.D.G.: Ulla och Gustaf af Uggla Foundation 2018-02670 https://staff.ki.se/ulla-and-gustaf-af-uggla-foundation L.M.D.G.: Reumatikerförbundet R-861801, R-932138 https://reumatiker.se L.M.D.G.: Konung Gustaf V:s 80-årsfond FAI-2018-0518, FAI-2019-0597 https://www.kungahuset.se/monarkinhovstaterna/kungligastiftelser/forskning/konunggustafvs80arsfond/ L.M.D.G.: Stiftelsen Professor Nanna Svartz Fond 2019-00318 https://www.stiftelsemedel.se/stiftelsen-professor-nanna-svartz-fond/ The funders had no role in study design, data collection and analysis, decision to publish, or preparation of the manuscript.

**Competing interests:** The authors have declared that no competing interests exist.

## Introduction

Genetic mapping studies of complex human traits have identified thousands of genetic loci implicated in the susceptibility to complex diseases [1, 2]. In other words, genome-wide association studies (GWAS) have linked thousands of single-nucleotide polymorphisms with complex human traits [1, 2]. These findings have been beneficial in areas such as drug repositioning [1, 3–5]. However, the identified individual genetic associations seldom exhibit strong disease risks and explain a small portion of the calculated heritability for each trait [1, 6–8]. This implies that the genetic associations have a very poor predictive value [6]. Comprehensive interaction analysis between, and among, risk factors is an important tool for understanding the genetic background of complex traits [7–11]. In general, interaction is said to be present when the combined effect magnitude of two or more factors is significantly different from the combined effect magnitude predicted by the model being tested [12–14]. Nevertheless, it is challenging to address whether, and how, genetic risk factors interact in shaping human traits, and to biologically interpret those interactions [9, 14]. Additionally, there is often confusion in the interpretation of the results from different interaction models. We therefore, in this study, provide a framework of simulated scenarios that represent common and simple processes where two or more genetic factors may interplay in disease causation. We investigate whether, and how, two or multiple genetic factors interact in each simulated scenario. We also address these relationships in real data from a case-control study in rheumatoid arthritis (RA), the Swedish epidemiological investigation of RA (EIRA) [15, 16].

The association between an individual genetic variant and an outcome (e.g., disease) is typically quantified as an odds ratio (OR) or relative risks (RR). The case-control design is often used for diseases with a low prevalence, where the resulting ORs approximate the RRs. Interactions studies generally examine two risk factors at a time, yielding three ORs (or RRs) notated as [13, 14]: $OR_{11}$ for carrying both risk factors; $OR_{10}$ and $OR_{01}$ for the exclusive combinations; and $OR_{00}$ for the absence of both risk factors, which is used as reference ($OR_{00} = 1$). Two different models are commonly used to test interaction, the additive (whose *null hypothesis* is $OR_{11} = OR_{10} + OR_{01} - 1$) and the multiplicative (whose *null hypothesis* is $OR_{11} = OR_{10} \times OR_{01}$). The additive model builds on the *sufficient cause* concept of KJ Rothman [12], who showed that if two factors are part of the same sufficient cause of a disease (e.g., pathway), then their joint risk will be larger than the *sum* thereof (often termed "departure from additivity"). This additive model has been criticized for always giving positive results when used as a null hypothesis [17, 18]. On the other hand, the multiplicative model has been criticized as a statistical convenience without a theoretical basis, boosted by the implicit multiplicativity in logistic regression [17, 19, 20]. There is often confusion regarding when and how to use and interpret each of these models.

In addition to evaluate the additive and multiplicative interaction models, we study a third model in this paper, the multifactorial threshold model [21]. This model assumes that there is a minimum number of factors required for disease causation, an assumption with a theoretical foundation in the concept of genetic liability [22]. Until now, no a statistical metric for the multifactorial threshold model has been available. We therefore propose a new metric, AMLES (additive-to-multiplicative linear effect scale), which compares the number of risk factors to a threshold value. Our proposed metric could help to determine the validity of the multifactorial threshold model, which has been criticized for simplifying disease biology [21].

Simulations are able to give answers with a high level of precision if enough data points are generated. They also provide great flexibility to fit a number of scenarios. We here build different scenarios in each of which we describe the cause of the outcome (disease or otherwise), with the purpose of tracing how a particular type of causality would be represented in an

interaction study. Our intension is to include commonly occurring and simple real-world scenarios, and practical examples from the literature are provided. We further build scenarios where we model situations with confounding factors commonly identified in genetic association studies, such as population stratification. We also show when the modelled scenarios can be distinguished in real data, and where they cannot. Across all simulations the risk factors are dichotomous, and neither necessary nor sufficient for disease to occur. While most of our simulated scenarios have two risk factors, we include modelling of more than two factors and illustrate how it is possible to elucidate multifactorial interactions from a set of pairwise interactions. We compare the simulated scenarios to additive and multiplicative risk scales, aiming to contribute to the understanding and interpretation of different models of interaction. Finally, we propose a metric in the framework of the multifactorial threshold model, which can estimate the fit to the additive and multiplicative models, as well as testing against a particular threshold level involving more than two factors.

## Methods

### Simulations

To better understand interactions between risk factors, we first set up simulations of causation, which are represented in Fig 1A to 1C. Each scenario included a dichotomous outcome (present/absent), and causative processes, termed components. For each of the different scenarios, we performed 1,000 simulations (Figs 1A–1C, 2B, 3A, 3B, S1A–S1C, S2A and S2B Figs). Each simulation consisted of one million data points (i.e., simulated individuals) where presence or absence of a risk allele and status of case or control was assigned in accordance with each model. As sensitivity analysis, we performed simulations with 100,000 data points. These simulations gave essentially the same results and were therefore not included. For the sake of simplicity, binary factors, corresponding to a dominant or recessive scenario in genetics, were used. We also used subgroups with equal ratio of cases and controls, named components, except for the three factors model (Fig 2B), where components 2 and 3 had the same ratio and component 1 had the same ratio as the logical-*OR* combination for the other two.

Allele frequencies for the two exposures, named X and Z, tested in each model, were established before each simulation by drawing a random value within a given range. For instance, if the lower frequency for the risk factor Z was set, a random value between 5% and 15% was chosen, then the higher frequency for Z was set by multiplying this figure by a random number between 1.1 and 4. The allele frequency for the exposure X was established in a similar fashion but using a range between 5% and 25% to select the random value, which was then multiplied by a random number between 1.1 and 2 to set the higher frequency.

For each scenario we calculated several metrics. We calculated ORs for the combinations of the risk factors: $OR_{01}$ for $X = 0$, $Z = 1$; $OR_{10}$ for $X = 1$, $Z = 0$; $OR_{11}$ for $X = 1$, $Z = 1$ (Fig 1D), to address interactions between two risk factors using the additive (with the null hypothesis $OR_{11} = OR_{10} + OR01 − 1$) and the multiplicative (with the null hypothesis $OR_{11} = OR_{10} \times OR_{01}$) models. Due to the nature of the components, more risk factors can exist in each scenario without interfering with the calculations. For each component there was one risk factor that was explicitly made to correlate with its "present" state (Fig 1A–1C), and the frequencies for the risk factor were varied between simulation runs. We subsampled the controls to match the number of cases and resemble how case-control studies reflect the population in a biased way (Fig 1). We calculated Pearson correlation coefficients between the risk factors X and Z (Fig 1E). For the versions of the scenarios without the subsampling (S1 Fig) we calculated RRs instead, as that is the appropriate population measure.

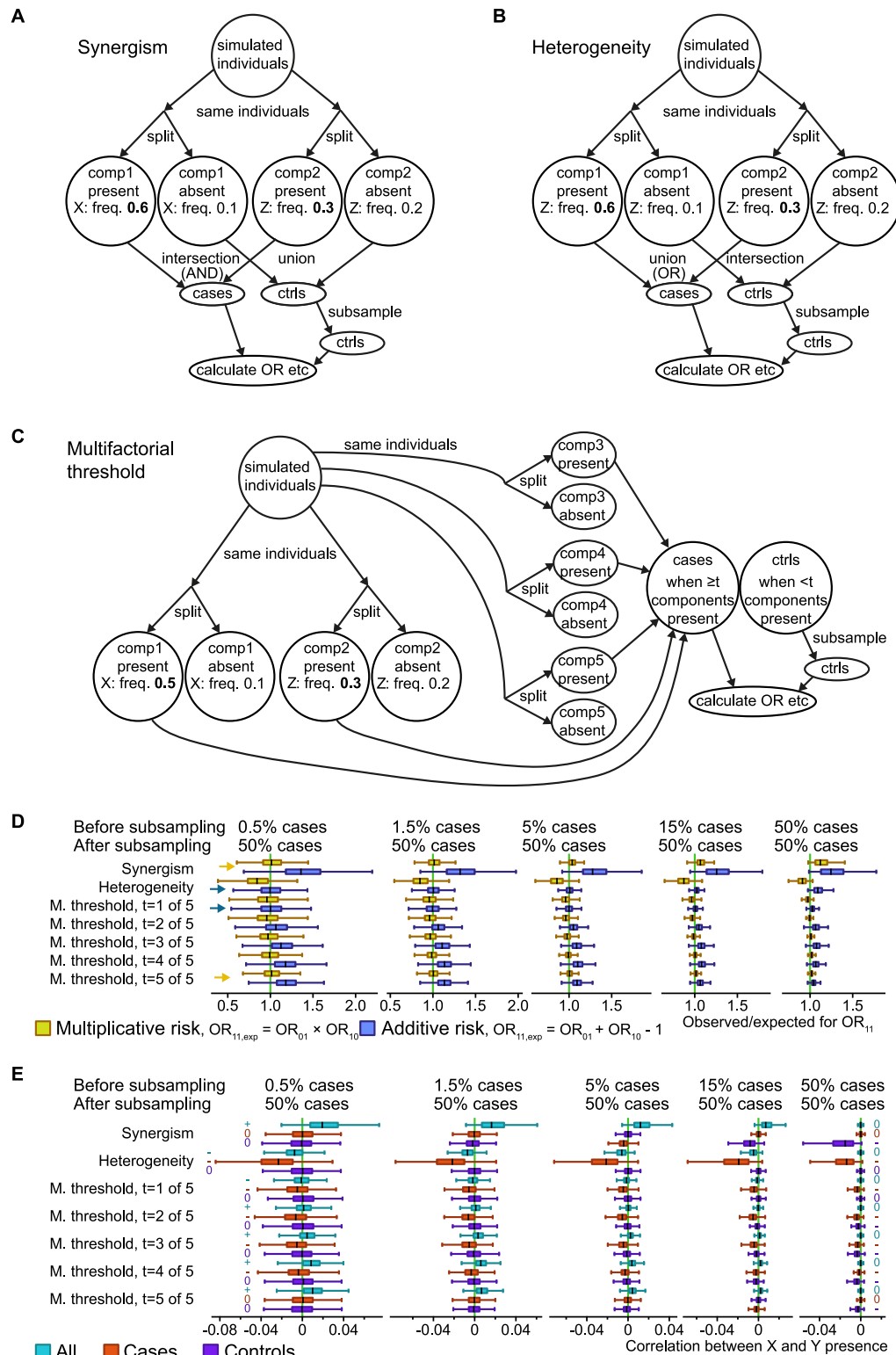

**Fig 1. Three simulated causal scenarios with selection of equal numbers of cases and controls. A-C:** Simulation schemes for three generalized scenarios in a case-control study context: Synergism of causes (A), heterogeneity of causes (B), and a multifactorial or 5-factor threshold (C). The numbers are example frequencies, and numbers in bold highlight the higher frequencies of the simulated risk factors (X and Z) associated with disease. For example, "Z: freq. 0.3" means that each simulated individual in the group had a 30% chance of being assigned the risk factor Z. Numbers in italic are

the average frequency in the other group of simulated individuals; note that this will depend on the prevalence (which is adjusted in the scenarios in the "split" into cases and controls). "Components" (comp1 and comp2) were used as a strategy to obtain probabilistic risk factors. **D**: ORs for double risk ($OR_{11}$) were calculated from the simulation scenarios, with boxes summarizing 1,000 simulation runs with different risk factor frequencies. The observed $OR_{11}$ were compared to the expected combinations of the odds ratio for single risk ($OR_{10}$ and $OR_{01}$) in the additive and multiplicative models. Boxplots show median and quartiles for the simulations, but extreme values are omitted for clarity. Yellow arrows highlight where the median is visibly close to the null hypothesis for the multiplicative model, while blue arrows do the same for the additive model, for the two most extreme simulated prevalence rates. "M. threshold" refers to multifactorial threshold (scenario C). **E**: Correlation coefficients between the risk factors X and Z, for three sample sets (all, cases only, controls only). The relevant signal in each case is whether the median is negative, zero or positive, highlighted with a -, 0 or + symbol for the two most extreme simulated prevalence rates.

## Interactions in rheumatoid arthritis GWAS

We evaluated both additive and multiplicative interaction models on a human case-control, genome-wide association dataset for anti-citrullinated protein antibody positive (ACPA-positive) rheumatoid arthritis (RA), from EIRA [15, 16]. The two top genetic risk factors for ACPA-positive RA in European-descendent populations, *HLA-DRB1* shared epitope alleles and *PTPN22* rs2476601 T, were tested against all non-*HLA* risk SNPs. Shared epitope (SE) is a group of *HLA-DRB1* alleles with similar effects, and rs2476601 is a non-synonymous coding variant of the *PTPN22* gene.

Genotyped and imputed GWAS data from the EIRA study were used in this part of the study (see [15] for sources included). Standard data filtering was performed as previously described [15]. Briefly, genotyping missing rate higher or equal to 5% and *P*-values of less than 0.001 for Hardy-Weinberg equilibrium in controls were excluded. The SNPs located in the extended HLA region (chr6:27339429–34586722, GRCh37/hg19) were removed, due to the high linkage disequilibrium and possible independent signals of association with ACPA-positive RA in the locus.

Departures from additivity or multiplicativity for risk factors were estimated pairwise in the imputed GWAS data (3,138,911 SNPs for the test with *HLA-DRB1* shared epitope and 3,308,784 SNPs for the test with *PTPN22* rs2476601 T), using GEISA [23], where a dominant model was assumed. In order to control by differences in allelic frequencies due to population stratification and sex, the first ten principal components (which summarized the genotyping data) and sex were used as covariates in this analysis. A cut-off of minimum five individuals for each OR combination was applied. The *HLA-DRB1* shared epitope alleles included *01 (except *0103), *0404, *0405 and *0408 and *1001. The *P*-values for interaction from these analyses are plotted in Fig 4A and 4B. We included only SNPs at risk allele frequencies between 10% and 50%; however, when we tested all the SNPs at a minor allele frequency above 1% the result provided the same conclusion.

To address both additive and multiplicative risk scales and evaluate the behavior of the ORs for double risk exposure ($OR_{11}$—Fig 4C, 4D, S3 and S4 Figs), we used genotyped EIRA GWAS data (281,195 SNPs). For this analysis, the data was transposed using Plink 1.07 [24]. Known risk SNPs were selected based on ORs higher than 1.1 and 95% confidence intervals do not overlapping 1, together with the criterion of having been reported as associated to RA in published case-control RA GWAS [5, 25, 26].

## Computational packages

Calculations were done using Python, including the packages NumPy [27], SciPy [28], Matplotlib [29], pandas [30], scikit-learn [31], seaborn [32], jupyter and geneview [33].

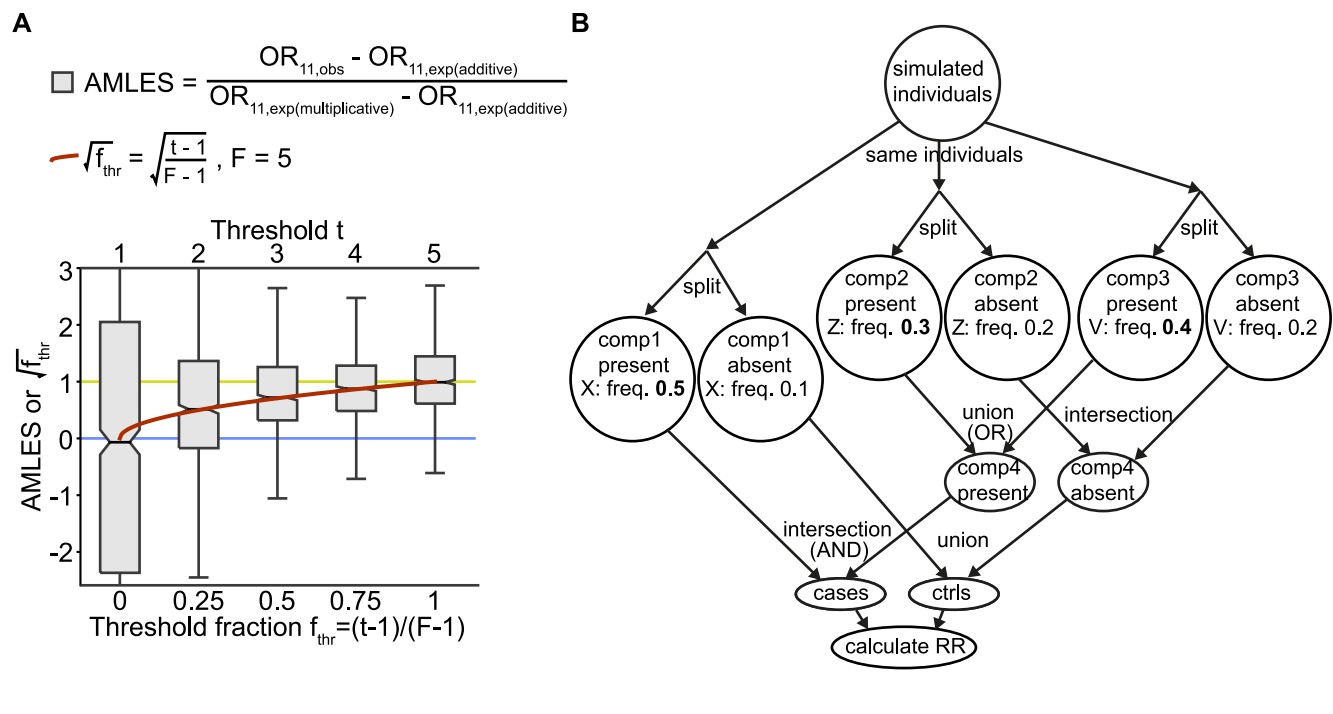

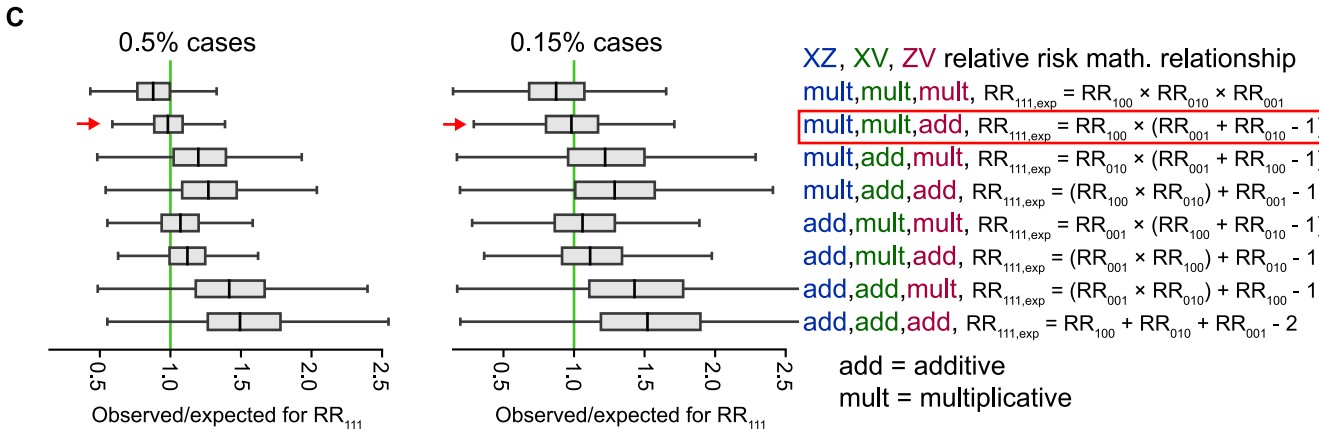

**Fig 2. Relationships among risk factors in the context of the multifactorial threshold model and a 3-factor causal scenario. A**: AMLES (additive-to-multiplicative linear effect scale; grey boxes) and $\sqrt{f_{est}}$ (red curve) calculated using the scenario in Fig 1C and plotted on the y axis. The notches in the box plots show bootstrapped 95% confidence intervals for the medians. The simulation had 0.5% cases and 99.5% controls, with a range ±0.04%. **B**: A 3-factor simulated scenario. The numbers are example frequencies, and frequencies in bold highlight the higher frequencies representing association with disease. For example, "Z: freq. 0.3" means that each simulated individual in the group had a 30% chance of being assigned the risk factor Z. This scenario required one "component" (where X is associated with risk) together with either one of another two components (where either Z or V is associated with risk) to produce the outcome. **C**: Comparison of the eight possible combinations of additive (add) and multiplicative (mult) relationships from the simulation in (B). The second combination, "mult,mult,add" is highlighted by a box since the theoretical expectation was met. There are AND relations between X and Z, X and V but an OR relationship between Z and V. Therefore, $RR_{100}$ (where only X is present) is multiplied by the addition of Z and V's relative risks, assuming that at a low trait prevalence the results from logical AND and OR follow the multiplicative and additive models, respectively.

## Data access

Interaction tables are available at Mendeley Data, doi:10.17632/63b47w6zgr.1 and can be viewed at https://data.mendeley.com/datasets/63b47w6zgr/1. Code is available at https://github.com/danielramskold/additive_risk_heterogeneity_multiplicative_risk_synergism2 where we also provide the code used to generate the figures. Pre-existing genetic data from the

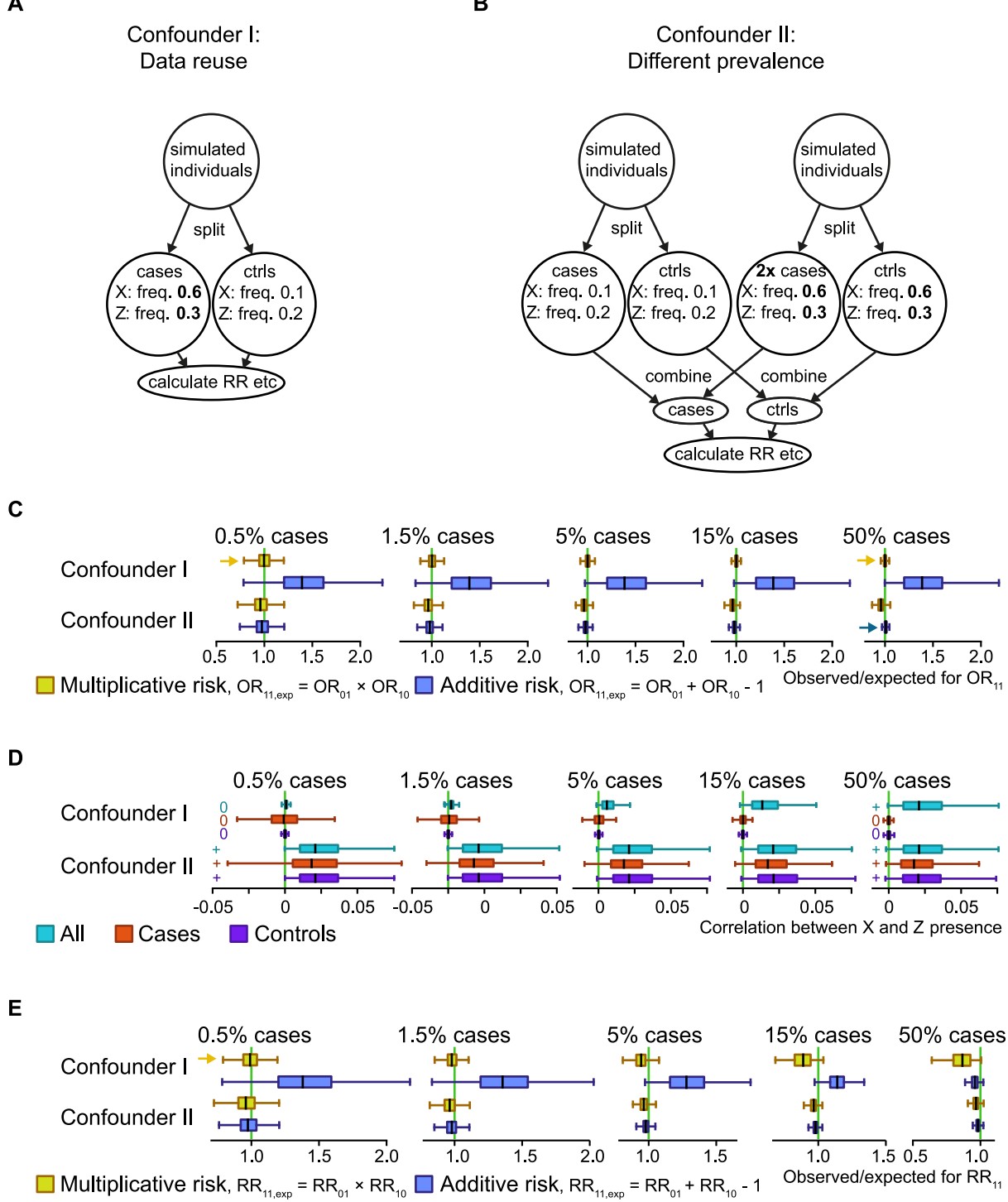

**Fig 3. Two simulated confounder scenarios. A**: Simulated confounder I, where the cases and controls come from different genetic backgrounds, which was simulated by testing interactions on the same data that the tested genetic risk factors were selected from. **B**: Simulated confounder II, where a mix of groups with both different genetic backgrounds and different prevalence rates were simulated. The numbers are example frequencies, and frequencies in bold highlight the higher frequencies of the simulated risk factors (X and Z) associated with disease. For example, "Z: freq. 0.3" means that each simulated individual in the group had a 30% chance of being assigned the risk factor Z. In each indicated circle the two risk factors have been added in an uncorrelated/independent manner. **C**: For each of the simulated confounder scenarios, the observed $OR_{11}$ (odds ratios for double risk) were compared to the expected ones for both the additive and multiplicative models. The boxes summarizing 1,000 simulation runs with different risk factor frequencies. The boxplots show median and quartiles for the

simulations, but extreme values are omitted for clarity. Yellow arrows highlight where the median is visibly close to the null hypothesis for the multiplicative model, while blue arrows do the same for the additive model, for the two most extreme simulated trait prevalences. **D**: Correlation coefficients between the risk factors X and Z. The relevant signal in each case is whether the median is negative, zero or positive, highlighted with a -, 0 or + symbol for the two most extreme simulated prevalence rates. **E**: Like (C), but for RR.

EIRA cohort was used. Due to ethical considerations, data from EIRA cannot be publicly shared. Please contact the principal investigators for data requests for applicable studies. For further information go to: https://www.eirasweden.se/Kontakt_EIRA.htm.

## Results

### Scenarios with risk factors contributing to the explicitly causal components

We arranged a simulation scenario for synergism, which is by far the most commonly identified type of interaction between genetic risk factors. For example, synergism has been shown between peptide-presenting *HLA-B* and the peptidase *ERAP1* in breast cancer [34]. We also

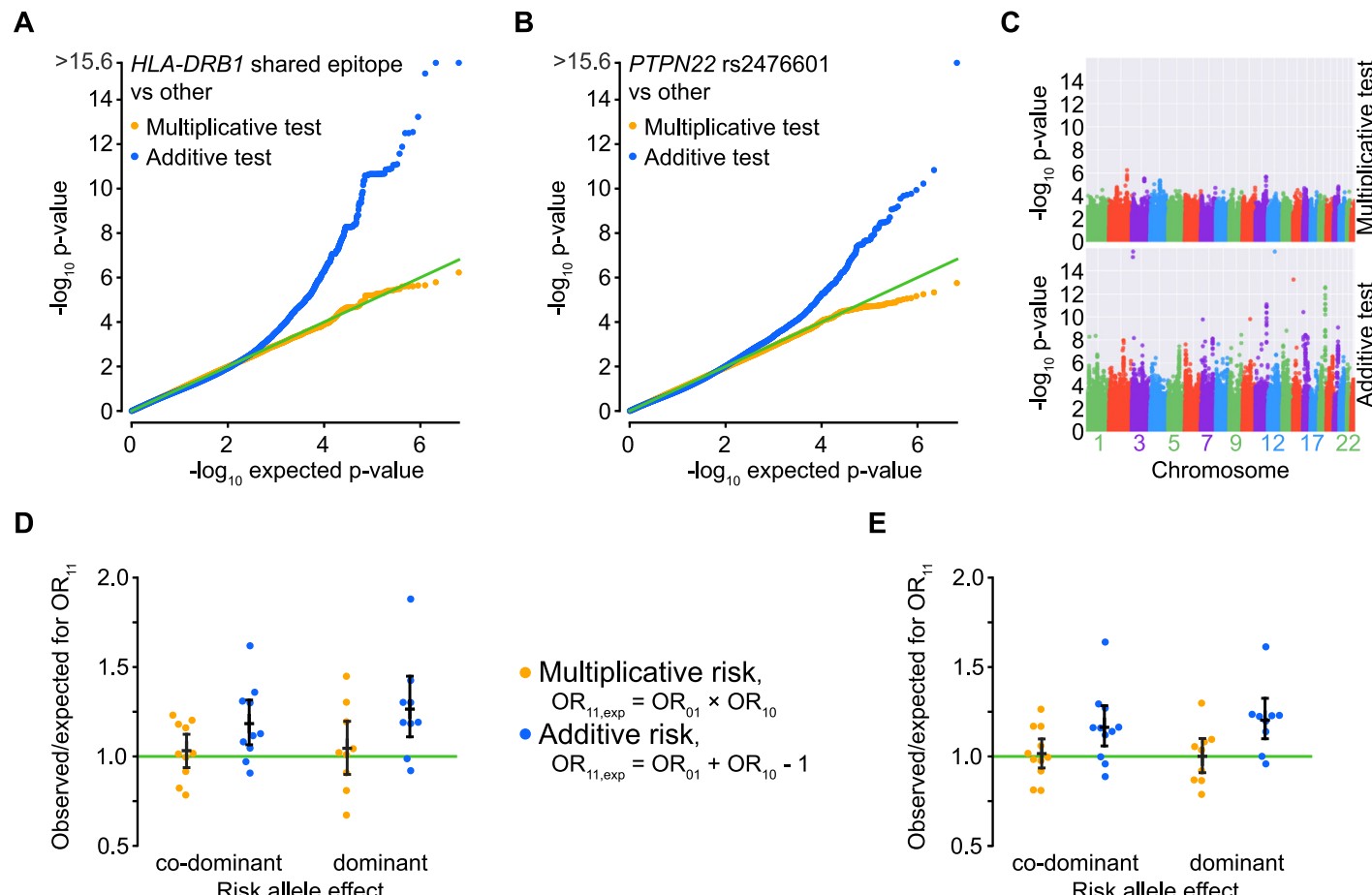

**Fig 4. Application to genome-wide association data for rheumatoid arthritis. A**-**B**: *P*-value distribution for two tests, one for deviation from additivity and one from multiplicativity. Two risk factors (see text for alleles) were tested in EIRA data against the rest of the genome except nearby SNPs. Each bin is 0.01 wide. A uniform distribution means a lack of deviation from the null model. **C**: Manhattan-like plots for the observed p-values from the tested multiplicativity or additivity models of interaction, for *HLA-DRB1* shared epitope against the rest of the genome except nearby SNPs. **D**: Odds ratio (OR) for *HLA-DRB1* shared epitope and one other risk factor, compared to that expected from an additive or multiplicative null model. Only known RA risk SNPs from the literature are shown. Black bars show median and 95% confidence intervals (bootstrap). **E**: The same SNPs as for D but shuffled within cases and shuffled within controls to match confounder I, thereby being a positive control for multiplicative ORs and allowing a comparison of dispersion with D.

simulated a scenario for the heterogeneity model, which is less commonly identified between genetic risk factors. An example of heterogeneity is the relationship between genetic variants from the *BLK* and *TNFSF4* genes in systemic lupus erythematosus [35]. In the synergism scenario, two components need to be in the "present" state for the outcome "case" to occur (i.e. logical AND; Fig 1A). In the heterogeneity scenario, it is enough that either one of two components are in the "present" state for a "case" outcome (i.e. logical OR; Fig 1B). We also sought to investigate the multifactorial threshold model, which has been thought of in terms of genetic liability [36] and has been exemplified in diabetes [21]. Here we chose five components, although any number from three upwards could have been used. In this model a minimum number of factors need to be present for the disease or trait to occur, which in our simulation set means that at least *t* number of components need to be in the "present" state for a "case" outcome (Fig 1C). We also present the results without subsampling, which matches population studies (S1 Fig). For the two risk factors, the synergism scenario corresponds to Rothman's synergism concept [13, 22], and the heterogeneity scenario corresponds to genetic heterogeneity.

For phenotypes with a low prevalence the presence of both risk factors had a multiplicative relation in the synergism scenario (Fig 1D), which is sometimes referred as "super-additive interaction", or "positive interaction" on the additive scale. For the heterogeneity scenario the presence of both risk factors instead had an additive relation at low prevalence (Fig 1D), which is sometimes referred as "negative interaction" in the multiplicative scale. The multifactorial threshold scenario equated to the heterogeneity scenario at threshold *t* = 1 and the synergism scenario at *t* = 5, where 5 meant *all* components needed to be present for phenotype to occur. We obtained intermediate results for intermediate thresholds; so that at low prevalence the joint behavior of the factors was above additive, but below multiplicative (Fig 1D). Without the subsampling (S1E Fig), the scenarios that produced multiplicative relationships lacked correlation between risk factors, both within cases, controls, and the two combined ("all"). However, the subsampling caused a positive correlation between risk factors in the combined ("all") group (Fig 1E). On the other hand, the additive and intermediate relations, without the subsampling, were reflected in negative correlations among risk factors in cases but not in controls (S1E Fig). This negative correlation between two risk factors can be understood theoretically in the context of the additive model of interaction, since two independent sufficient factors (meaning that there is heterogeneity of causation) should have a strongly negative correlation. Thus, a similar, but attenuated, pattern of correlation between non-sufficient risk factors is expected. We also present results from a simulation of unbiased random sampling (without the subsampling step) that is equivalent to population studies. In this analysis, we detected a multiplicative relation between RRs or a synergism scenario at every prevalence rate (fraction of cases in the population). For the heterogeneity scenario, the relationship between risk factors was additive at low prevalence but less-than-additive at high prevalence.

## AMLES, an interaction metric with a simple interpretation for multifactorial thresholds

As we have seen, the multifactorial threshold model gives rise to intermediary interaction relationships between additivity and multiplicativity for the combinations of the risk factors involved (Fig 1D). We therefore set a scale to calculate the intermediary relationships among risk factors. To anchor the scale at 0 for additive relationships, the expected effect from the additive model is subtracted from the observed double exposure OR. Consequently, to set the scale at 1 for multiplicative relationships, the previous subtraction is divided by the difference between the expected OR for the multiplicative model and the expected OR for the additive

model. This provides an additive-to-multiplicative linear scale (AMLES):

$$\text{AMLES} = (\text{OR}_{11}(\text{observed}) - \text{expected}(\text{additive}))/(\text{expected}(\text{multiplicative}) - \text{expected}(\text{additive}))$$

where $\text{expected}(\text{additive}) = \text{OR}_{10} + \text{OR}_{01} - 1$ and $\text{expected}(\text{multiplicative}) = \text{OR}_{10} \times \text{OR}_{01}$.

Then, to be able to compare AMLES with the threshold in our simulated threshold model, we scaled the threshold as the ratio:

$$f_{\text{thr}} = (t - 1)/(F - 1)$$

where $F$ is the number of components (factors) that could cross the threshold $t$, in order to have a 0 to 1 scale (see Fig 2). We found that the square of the threshold ratio fits the scaled OR. This means that the OR at a low prevalence (such as 0.5%) for doubly exposed ($\text{OR}_{11}$) was

$$(1 - \sqrt{f}) \times \text{expected}(\text{additive}) + \sqrt{f} \times \text{expected}(\text{multiplicative}),$$

where $\sqrt{f}$ is the square root of $f_{\text{thr}}$. The metric used, AMLES has the convenient properties of having both a natural lower value (0 for additive) and higher value (1 for multiplicative), as well as an interpretable scale between these values, therefore it is also related to the threshold fraction $f_{\text{thr}}$. However, the metric $f_{\text{est}} = \text{sign}(\text{AMLES}) \times (\text{AMLES})^2$ has a more natural scale, but unlike AMLES this is not symmetric around the median, making it a less convenient scale (S2 Fig). AMLES is related to another measure of interaction size, relative excess risk due to interaction ($\text{RERI} = \text{RR}_{11} - \text{RR}_{10} - \text{RR}_{01} + 1$) [12, 22], by $\text{AMLES} = \text{RERI} / (\text{OR}_{10} - 1) / (\text{OR}_{01} - 1)$ in cases when ORs and RRs tend to be the same. AMLES could perhaps be used instead of measures like attributable proportion, synergy index and RERI, given that it can be indicative of multiplicative or additive risk regardless of the scale (additive or multiplicative) used for testing.

## Three risk factors

Relationships involving more than two potential causes can be analyzed by pair-wise interactions, as can be seen in our simulation scenario based on the multifactorial threshold model. Although an outcome with more than two causes will not always entail a threshold model, it may be possible to break down the multifactorial relationships into pairs by those involving causal heterogeneity and synergism. For example, in the case of outcome = $X$ AND ($Z$ OR $V$) = ($X$ AND $Z$) OR ($X$ AND $V$), the relationships among these risk factors should be distinguishable as causal heterogeneity between $Z$ and $V$ (from the logical OR) and synergism between $X$ and each of the other risk factors (from the logical AND) (Fig 2B). This might be how body mass index (BMI) interacts with variants of the *F5* and *ABO* genes (as if X = BMI, Z = *F5* SNP, V = *ABO* SNP) in ventral thromboembolism risk [37, 38]. Our simulation performs as predicted when we tested this example against the expected RR from all eight possible combinations of both logical OR and AND (Fig 2C, S1C Fig).

## Scenarios where confounder effects cause false interaction

We investigated two confounder scenarios, neither of which include any causal relationship between the risk factors (e.g., heterogeneity or synergism) and are therefore applicable when either or both are false risk factors. The first scenario describes when the genetic background for cases and controls are mismatched (Fig 3A). It can also be described as an effect of data reuse, where the same data set is used to define risk factors and to calculate interactions, as was done in a previous interaction study [18]. In the other scenario we simulated multiple ancestry groups, which may translate to different allelic frequencies across groups, a well-known

confounder that is generally considered in the design or corrected for in genetic association studies [39]. In our simulation we condensed it down to two groups (Fig 3B). We did not model linkage disequilibrium (LD) in our simulations, as the type of simulations we used do not accommodate LD, but we tested our observations on a real GWAS of ACPA-positive RA (anti-citrullinated protein antibody positive rheumatoid arthritis). It may be advisable to prune for LD when addressing interactions in genome-wide data sets [40].

For the first confounder scenario (Fig 3A), ORs (Fig 3C) and correlations (Fig 3D) at high fractions of cases are more relevant than RRs (Fig 3E). We found the ORs relationship to be multiplicative, regardless of the balance between cases and controls, which makes this confounder scenario indistinguishable from real synergism. This was also true for correlation coefficients, where it would only be in population data that it would be possible to distinguish synergism from the confounder scenario (Figs 1E and 3D, S1E Fig) by the former's lack of correlation in that situation. The second confounder scenario (Fig 3B) had negative-additive ORs and RRs relationships, matching additive in a high proportion of cases (Fig 3C and 3E). The risk factors correlation among the three groups used (all, cases, controls) was always positive (Fig 3D), which should be useful in distinguishing this confounder scenario from real interactions. We also tested what happens when adding a biased subsampling step, as in Fig 1A–1C, to Confounder II, but it made no difference to the results (S3 Fig). We also devised one simulation scheme that always produced additive RRs relationships and one that always produced additive ORs relationships (S4 Fig), because such schemes could guide randomization, and to complement the always-multiplicative OR of the first confounder scenario.

## Example from rheumatoid arthritis

Both the synergism and heterogeneity scenarios represent interesting relationships between two risk factors, thus the appropriate interaction model to use depends on the hypothesis one is interested in. Alternatively, a hypothesis-free approach would be ideal, and the closest to that is to evaluate both additive and multiplicative hypotheses, as this would cover both models as well as threshold-based scenarios due to their intermediate nature (i.e. they would fail both types of tests, in the positive direction for additive and negative direction for multiplicative). We therefore evaluated both additive and multiplicative interaction for the two top genetic risk factors (*HLA-DRB1* shared epitope alleles and *PTPN22* rs2476601 T allele) for ACPA-positive RA in European-descent population, using GWAS data from the EIRA study. We detected no deviation for multiplicativity, but did for additivity (Fig 4A and 4B), as reported before for *HLA-DRB1* [15]. The new simulations presented here increases our ability to interpret this result as a widespread interaction between *HLA-DRB1* shared epitope and all non-*HLA* genetic risk factors, in the common meaning of interaction where synergism is a type of interaction. Correlation analyses backed up synergism (but not heterogeneity or population stratification) as an appropriate interpretation of the results (S5 Fig). From this, we could derive that the *HLA-DRB1* shared epitope cannot be substituted or phenocopied by a non-*HLA* genetic risk factor for its part in the chain of ACPA-positive RA etiology (Fig 4A, 4C and 4D). The same was the case for the *PTPN22* risk allele, given the similarities in *P*-value distributions observed (Fig 4A and 4B). For both set of tests there were a majority of tested loci where there was too little data to distinguish additive from multiplicative ORs. We followed up the results of multiplicativity by looking only at known risk SNPs (Fig 4D), finding results similar to a randomization based on the Confounder I scenario (and therefore bound to produce multiplicative odds ratios), with similar variability ($P = 0.6$–$0.8$, Levene's test, $n = 9$–$11$ SNPs) implying a dearth of non-multiplicative ORs relationships (Fig 4E). This randomization is the same as Test III of a previous study by Ignac *et al* [41]. We also devised a randomization

scheme creating additive ORs based on the Additive O scheme of S4 Fig (intended to provide always additive effect relationships) and tested it on the full SNP set (S6 Fig). This result showed a very noticeable deviation from the real data, as expected.

## Discussion

We herein present a simulation approach intended to help with the interpretation of additive and multiplicative models of interaction of RRs and ORs. We show that addition of ORs, or negative deviation from the multiplicative interaction model ($OR_{11} < 1$, when testing $OR_{11} = OR_{10} \times OR_{01}$), occurs when the risk comprises two independent risk factors (heterogeneity scenario, Fig 1B), or a process that approximating to that setup at a given fraction of cases (S4 Fig). Multiplication of ORs, or positive deviation from additive interaction model ($OR_{11} > 1$, when testing $OR_{11} = OR_{10} + OR_{01} - 1$), follows if two different mechanisms are required for disease (synergism scenario, Fig 1A). Depending on the hypothesis one should therefore choose the appropriate statistical test. To identify deviation from synergism, deviation from a multiplicative relationship could be tested. Often however, if we are interested in testing whether disease is caused by the interaction of two factors, it is appropriate to test for deviation from additivity. A multiplicative assumption would have merit in our testing against *HLA-DRB1* shared epitope and *PTPN22*, if ACPA-positive RA were a homogeneous set of causes, rather than the kind of heterogeneity of causation that we have shown gives rise to additivity between risk factors. Nevertheless, ACPA-positive RA may have a certain level of heterogeneity of causes, despite being defined as a subgroup of patients where ACPA positivity is a mediating risk factor [42]. Therefore, in this study, we also tested the multiplicative inter-action between the strongest genetic risk for ACPA-positive RA and other risk-SNPs in the same material as previously published by Diaz-Gallo *et al* [15], these results showed that there is not deviation from the null hypothesis for the multiplicative model of interaction. However, we only tested the main other model, not for example some version of the multifactorial threshold model. In light of the new understanding that our simulations give, the presence of deviation from additivity, along with no deviation from multiplicativity, supports the existence of widespread synergism between the genetic risk factors in causing ACPA-positive RA. This result is also applicable to the estimation of heritability for RA, since the assumption of an additive model would lead to an underestimation of narrow-sense heritability [43].

Synergism can be viewed as a chain of events scenario, whereas causal heterogeneity corresponds to phenocopying. In terms of Rothman's sufficient-cause model, the risk factors X and Z in the synergism scenario correspond to risk factors in the same cause, referred to as causal co-action, joint action or synergism, whereas X and Z in the heterogeneity scenario correspond to risk factors in different causes [22].

The fact that most loci showed no statistically significant deviation from neither additive nor multiplicative interaction will be the unfortunate reality for many applications of interaction testing. While statistical power for single risk factor testing scales with the inverse square of the number of samples, already forcing large GWAS sample sizes, the statistical power for interaction testing scales to the inverse power of four [43], thus requiring far larger sample sizes than standard association testing.

Our inspiration for this work came from a simulation study [18] in which case or control status was randomly assigned, and one risk factor was simulated to resemble the strongest genetic risk factor for ACPA-positive RA, and interaction with other risk factors (selected by *P*-value for risk) was computed. The simulation led to an overrepresentation of additive inter-actions (i.e., deviation for additive odds ratios). Our Confounder I scenario (Fig 3A) was produced an analogous result. The author [18] noted that his simulation produced a

multiplicative null model that does not match additivity and concluded that the additive interactions observed were erroneous, as no interaction should be present. We propose an alternative interpretation, based on the convergences we found: the Confounder I scenario, and thereby also the previous simulation [18] perfectly mimics synergism in a case-control study with a similar number of cases and controls selected from a population with a low prevalence, as can be seen by comparing our figures (Figs 1D, 1E, 3C and 3D). By low prevalence, we mean the 0.5% mark in our simulations, which is similar to RA prevalence: 0.7% for all RA in Sweden [44], of which 60% are ACPA-positive [45]). It should be no surprise that a simulation [18] of synergism, which of course is interaction [46], produces many significant results, although it is of course worrying that a confounder like data reuse can produce this effect, as the study [18] showed. We conclude that the previous simulation study [18] is in line with the confusion over additive and multiplicative interaction that can sometimes be found in the literature [47], highlighting the need to understand better the relationships among risk factors that different interaction types imply.

In this simulation study we demonstrate the causal interpretations of additive and multiplicative interaction in both the RR and OR settings. Some of this has been understood intuitively in the past, especially the connection between multiplicative effect and logical AND [48], but here we contribute with further clarification of the situation through simulation. We hope that this will help guide the interpretation of future interaction studies.

## Supporting information

**S1 Fig. Three simulated causal scenarios.** **A**-**C**: Simulation schemes for three generalized scenarios: synergism of causes (A), heterogeneity of causes (B) and a 5-factor threshold (C). The numbers are example frequencies, and frequencies in bold highlight the higher frequencies of the simulated risk factors (X and Z) associated with disease. For example, "Z: freq. 0.3" means that each simulated individual in the group had a 30% chance of being assigned the risk factor Z. The numbers in italics are the average frequency in the other group of simulated individuals; note that this will depend on the prevalence (which is adjusted in the scenarios in the split between cases and controls). "Components" (comp1 and comp2) were used as a strategy to obtain probabilistic risk factors. **D**: The relative risks for double risk ($RR_{11}$) calculated from the simulation scenarios, with boxes summarizing 1,000 simulation runs with different risk factor frequencies. The observed $RR_{11}$ are compared to the additive and multiplicative combinations of the relative risks for single risk ($RR_{10}$ and $RR_{01}$). Boxplots show median and quartiles for the simulations, but extreme values are omitted for clarity. Yellow arrows highlight where the median is visibly close to multiplicativity, while blue arrows do the same for additivity, for the two most extreme simulated prevalence rates. "M. threshold" means scenario C. **E**: Correlation coefficients between the risk factors X and Z. The relevant signal in each case is whether the median is negative, zero or positive, highlighted with a -, 0 or + symbol for the two most extreme simulated prevalence rates. We left out the highest prevalence results due to division-with-zero difficulties in the multiple threshold scenario.
(PDF)

**S2 Fig. Multifactorial threshold model with $f_{est}$ metric.** The y axis is the signed square of the y axis in Fig 2A, otherwise this is the same plot, i.e. based on the scenario in Fig 1C. The red curve is the threshold fraction; this and $f_{est}$ make up the y axis. The notches in the box plots show bootstrapped 95% confidence intervals for the medians. The simulation had 0.5% cases and the rest controls, with a range ±0.04%.
(PDF)

**S3 Fig. No apparent difference from subsampling to equal number of controls and cases in confounder II.** As in the scheme to the left, we simulated with (purple arrows) or without (green arrow) a step that reduced the number of controls to the same as the number of cases. As can be seen in the box plots above, this does not appear to have any effect on the results. (PDF)

**S4 Fig. Schemes providing additivity. A-B**: Simulation for two schemes intended to produce additive effect. The numbers are example frequencies, and frequencies in bold highlight the higher frequencies of the simulated risk factors (X and Z) associated with disease. For example, "Z: freq. 0.3" means that each simulated individual in the group had a 30% chance of being assigned the risk factor Z. The numbers in italics are the average frequency in the other group of simulated individuals; note that this will depend on how many cases there are to controls, specifically, higher_freq × case_fraction + lower_freq × (1 –case_fraction). In each indicated circle the two risk factors have been added in an uncorrelated/independent manner. **C-D**: The relative risks (C) and odds ratios (D) for double risk ($RR_{11}$ or $OR_{11}$) calculated from the simulation schemes, with boxes summarizing 1,000 simulation runs with different risk factor frequencies. The observed $RR_{11}$ were compared to the expected combinations of the relative risks for single risk ($OR_{10}$ and $OR_{01}$) in the additive and multiplicative models. Boxplots show median and quartiles for the simulations, but extreme values are omitted for clarity. Yellow arrows highlight where the median is visibly close to the null hypothesis for the multiplicative model, while blue arrows do the same for the additive model, for the two most extreme simulated trait prevalence rates. **E**: Correlation coefficients between the risk factors X and Z. The relevant signal in each case is whether the median is negative, zero or positive, highlighted with a -, 0 or + symbol for the two most extreme simulated prevalence rates. (PDF)

**S5 Fig. Shared epitope correlation relationships.** Correlation between *HLA-DRB1* Shared Epitope (calculated as codominant) and other SNPs (calculated as dominant) in EIRA data within cases, control or all samples, for all non-HLA SNPs (A, C) or only known non-HLA risk SNPs (B). P-values come from 1-sample t-tests against zero. Regressing out sex and the first ten principal components gave essentially the same result (C) as without this step (A, B). (PDF)

**S6 Fig. Comparison of EIRA double risk odds ratios to two types of randomizations. A-C**: SE is *HLA-DRB1* Shared Epitope, "other" are non-*HLA* risk SNPs for ACPA-positive RA. Unmod. or Unmodified is the EIRA data as it is, Rand1 shuffles, for each locus independently, within cases and shuffling within groups to match the independence part of the confounder scenario of Fig 3A (confounder I), Rand2 assigns individuals randomly into two groups and then in one group samples risk factor SE from controls and in the other group samples the other SNP risk factor from controls, to match S2B Fig (Additive O). We suspect that Rand2 unfortunately moderates odds ratios, meaning it could use a better replacement. Unmodified and Rand1 have similar variability (P = 0.04–0.8, Levene's test, excluding (C) where calculations failed (NaN)). The middle of the scale for (C) is magnified to ease viewing. Blue arrows highlight where the median is visibly close to additivity, while yellow arrows do the same for multiplicativity. **D**: SNPs from Fig 4C and 4D, in varying numbers due to division-by-zero errors, with effect scaled between additive (blue line) and multiplicative (yellow line), as in Fig 2A. (PDF)

## Acknowledgments

We thank Anton Larsson for helping with the literature search, Lars Klareskog, Lars Alfreds-son and Leonid Padyukov for providing access to EIRA data and engaging in scientific discussions, and Janet Ahlberg for English editing.

## Author Contributions

**Conceptualization:** Boel Brynedal, Daniel Ramsköld.

**Formal analysis:** Lina-Marcela Diaz-Gallo, Daniel Ramsköld.

**Methodology:** Helga Westerlind, Daniel Ramsköld.

**Resources:** Lina-Marcela Diaz-Gallo.

**Software:** Daniel Ramsköld.

**Supervision:** Lina-Marcela Diaz-Gallo, Rickard Sandberg.

**Visualization:** Daniel Ramsköld.

**Writing – original draft:** Daniel Ramsköld.

**Writing – review & editing:** Lina-Marcela Diaz-Gallo, Boel Brynedal, Helga Westerlind, Rickard Sandberg, Daniel Ramsköld.

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
