## [Decision Letter · Decision Letter 0]

29 Oct 2020

PONE-D-20-24738

Understanding interactions between risk factors, and assessing the utility of the additive and multiplicative models through simulations

PLOS ONE

Dear Dr. Ramsköld,

Thank you for submitting your manuscript to PLOS ONE. After careful consideration, we feel that it has merit but does not fully meet PLOS ONE’s publication criteria as it currently stands. Therefore, we invite you to submit a revised version of the manuscript that addresses the points raised during the review process.

A **rebuttal letter** that responds to **EACH** point raised by the academic editor and reviewer(s). You should upload this letter as a separate file labeled 'Response to Reviewers'.A **marked-up copy** of your manuscript that highlights changes made to the original version. You should upload this as a separate file labeled 'Revised Manuscript with Track Changes'.An **unmarked version** of your revised paper without tracked changes. You should upload this as a separate file labeled 'Manuscript'.

We look forward to receiving your revised manuscript.

Kind regards,

Brecht Devleesschauwer

Academic Editor

PLOS ONE

Additional Editor Comments:

Based on the reviewer's comment, we invite the authors to submit a revised version of the manuscript with substantial revision in terms of clearly presenting the goals and the biological reasoning behind each presented simulation scenario.

In your revision note, please include EACH of the reviewer comments, provide your reply, and when relevant, include the modified/new text (or motivate why you decided not to modify the text). Note that failure to do so may result in a rejection of the manuscript.

Journal Requirements:

2. PLOS ONE publication criteria and journal policy require authors to make data underlying the findings described in their manuscript fully available (https://journals.plos.org/plosone/s/data-availability#loc-acceptable-data-access-restrictions). Please describe how other researchers could access the cohort datasets used in your study.

Reviewers' comments:

Reviewer's Responses to Questions

**Comments to the Author**

1. Is the manuscript technically sound, and do the data support the conclusions?

Reviewer #1: Yes

2. Has the statistical analysis been performed appropriately and rigorously? 

Reviewer #1: Yes

3. Have the authors made all data underlying the findings in their manuscript fully available?

Reviewer #1: Yes

4. Is the manuscript presented in an intelligible fashion and written in standard English?

Reviewer #1: Yes

5. Review Comments to the Author

Reviewer #1: This is a nice paper investigating interaction in several context and providing interpretation and advantages of measures of interaction. The research question is interesting, however I feel several points need to be clarified.

First, it is not fully clear what the practical utility of the study is. As it stands, this looks like a nice computational study without clear implications. It would be very useful to motivate biologically the simulation scenarios. How were they chosen? What possible situations do they represent? This would be really useful to understand the practical utility of the study beyond the presented example. I recommend providing some practical examples for each of the presented scenario, like situations where it could arise, or published papers. Also, and importantly, the introduction is built on introducing the need of evaluating interactions between genetics and environmental factors, but it seems that all simulations, as well as the provided example, are based on a situation of genetic-genetic interactions? I suggest either revise the introduction accordingly, or include more simulations and examples showing the utility of the measure in the context of GxE interactions.

Second, the abstract needs a substantial revision. It should stand by itself and convey the main messages of the paper, but now it is possible to understand the abstract only after having read the manuscript. Several concept are introduced without any explanation, like multifactorial models, threshold models, intermediaries between additive and multiplicative interaction. Simulations are not motivated. The sentence at line 6-7, which seems to be key, is not clear (missing a verb?)

Third, the newly introduced formula (lines 173 and after) seems to me the most useful and important contribution. However this (meaning the need of this formula and the potential advantages) are not sufficiently motivated in the introduction section. Given the importance of this results I suggest revising the introduction and the abstract to emphasize the need of an intermediary formula between additive and multiplicative interaction, as well as the potential advantages to motivate its introduction and use in different settings.

Minor comments.

line 7-8. For synergy the two risk factors are required, rather than contribute.

Lines 25-26. Since until here you have been talking about multiple risk factors, it is good to mention that you are now referring to only 2 factors.

Page 3. Y is universally used to refer to the outcome, I suggest using Z-X for referring to the 2 simultaneous exposures, and then another letter for the third exposure later.

Line 99. Principal components summarizing what?

Line 433. I couldn’t find detailed figure legends. I think these should be useful, to present what the different panels are showing and provide a brief understanding of the figures without the need of reading the full manuscript

6. PLOS authors have the option to publish the peer review history of their article (what does this mean?). If published, this will include your full peer review and any attached files.

Reviewer #1: No

---

## [Author Response · Author response to Decision Letter 0]

12 Dec 2020

This is a copy of the included rebuttal letter file (labeled Response to Reviewers), which has better formatting:

Editorial comments:

1. Please ensure that your manuscript meets PLOS ONE's style requirements, including those for file naming. The PLOSONE style templates can be found at

Based on the PLOS ONE’s style requirements, we have fixed the bolding of supplemental figure captions and removed the zip codes in the affiliations.

2. PLOS ONE publication criteria and journal policy require authors to make data underlying the findings described in their manuscript fully available (https://journals.plos.org/plosone/s/data-availability#loc-acceptable-data-access-restrictions). Please describe how other researchers could access the cohort datasets used in your study.

We have added text about the cohort dataset under Data Access, stating that "[d]ue to ethical permission, data from EIRA cannot be publicly shared. Please contact the principal investigators for data requests for applicable studies. For further information go to: https://www.eirasweden.se/Kontakt_EIRA.htm".

We will provide a DOI after acceptance, before publication.

We have now added the supplemental figure S3. Therefore, we removed the "data not shown" statement from the sentence at current line number 283.

We have also removed the "data not shown" from figure legend S6 and weakened the statement around it so that it does not require the data in question. The data itself can be found in eira_randomised_distibutions.ipynb in the paper's associated github page, but we choose not link it in the legend as it is not a core part of the research we present.

This is how S6 figure legend now reads:

"(A-C) SE is HLA-DRB1 shared epitope, “other” are non-HLA risk SNPs for ACPA-positive RA. Unmod. or Unmodified is the EIRA data as it is, Rand1 shuffles, for each locus independently, within cases and shuffling within groups to match the independence part of the confounder scenario of Fig 3A (Confounder I), Rand2 assigns individuals randomly into two groups and then in one group samples risk factor SE from controls and in the other group samples the other SNP risk factor from controls, to match S2B Fig (Additive O). We suspect that Rand2 unfortunately moderates odds ratios, meaning it could use a better replacement. Unmodified and Rand1 have similar variability (P=0.04-0.8, Levene’s test, excluding (C) where calculations failed (NaN)). The middle of the scale for (C) is magnified to ease viewing. Blue arrows highlight where the median is visibly close to additivity, while yellow arrows do the same for multiplicativity. (D) Genes from Figure 4C-D, in varying numbers due to division-by-zero errors, with effect scaled between additive (blue line) and multiplicative (yellow line), as in Fig 2A."

This is how the new supplementary figure looks:

Reviewers' comments:

Reviewer #1: This is a nice paper investigating interaction in several context and providing interpretation and advantages of measures of interaction. The research question is interesting, however I feel several points need to be clarified.

We thank the reviewer for her/his positive evaluation regarding the overall aim of our study. We have followed her/his suggestions to improve the way we are presenting the message of our investigation in the manuscript. The details of this are presented below.

First, it is not fully clear what the practical utility of the study is. As it stands, this looks like a nice computational study without clear implications. It would be very useful to motivate biologically the simulation scenarios. How were they chosen? What possible situations do they represent? This would be really useful to understand the practical utility of the study beyond the presented example. I recommend providing some practical examples for each of the presented scenario, like situations where it could arise, or published papers. 

They were chosen from two perspectives: 1) what does it mean to have an additive or multiplicative effect? and 2) what would data from the most common scenarios in the literature look like? It is indeed helpful to anchor the scenarios in the literature, and we have added practical examples from the literature to several scenarios: synergism, heterogeneity model, multifactorial threshold model and the three factor Fig 2B scenario. We also add a paper discussing and dealing with genetic background confounder (confounder II). We already reference a paper discussing the data reuse confounder (confounder I). The papers for the confounder scenarios are not practical examples per se, as papers are not as a rule published with known uncorrected confounders, but referencing the discussion about these scenarios is indeed useful. The additive scenarios of S4 Fig are not as far as we know connected to any practical situations, which is why they're in the supplement, although they might be useful for randomization or understanding additivity as we demonstrate in a tentative example in S6 Fig.

Also, and importantly, the introduction is built on introducing the need of evaluating interactions between genetics and environmental factors, but it seems that all simulations, as well as the provided example, are based on a situation of genetic-genetic interactions? I suggest either revise the introduction accordingly, or include more simulations and examples showing the utility of the measure in the context of GxE interactions.

We have revised the introduction to avoid any mention of environment and to specify that the paper discusses genetic interactions. 

Second, the abstract needs a substantial revision. It should stand by itself and convey the main messages of the paper, but now it is possible to understand the abstract only after having read the manuscript. Several concept are introduced without any explanation, like multifactorial models, threshold models, intermediaries between additive and multiplicative interaction. Simulations are not motivated. The sentence at line 6-7, which seems to be key, is not clear (missing a verb?)

We have rewritten the abstract.

Here is the new abstract:

"To understand the genetic background of complex diseases it is necessary to study beyond univariate associations. Therefore, interaction assessments of risk factors are important to uncover which, whether and how genetic risk variants act together on disease development. The principle of interaction analysis is to explore the combined effect’s magnitude of risk factors on disease causation. In this study, we investigate using simulations, different scenarios of causation to show whether and how the magnitude of the effect of two risk factors affect each other. We mainly but not exclusively address the two most commonly used interaction models, the additive and multiplicative, since there is often confusion regarding their use and interpretation. Our results show that when two risk factors are involved in the same chain of events, an interaction called synergism, their effect is multiplicative. Synergism is often described as deviation from additivity, which is a less precise allusion. Our results corroborate that it is often relevant to estimate additive effect relationships because they correspond to independent risk factors at low disease prevalence. Importantly, we also evaluate what is the threshold of more than two required risk factors for disease causation, called the multifactorial threshold model. We found a square root relationship between the threshold and an additive-to-multiplicative linear effect scale (AMLES), which ranges from 0 (equivalent to additive) to 1 (equivalent to multiplicative). Therefore, we propose AMLES as a metric that might be used to test different effects relationships at once, given that it can pinpoint multiplicative risk effects relationships in testing on the additive scale and vice versa. Finally, the utility of our simulation study was demonstrated in real data by analyzing and interpreting gene-gene odds ratios from a rheumatoid arthritis case-control cohort."

Third, the newly introduced formula (lines 173 and after) seems to me the most useful and important contribution. However this (meaning the need of this formula and the potential advantages) are not sufficiently motivated in the introduction section. Given the importance of this results I suggest revising the introduction and the abstract to emphasize the need of an intermediary formula between additive and multiplicative interaction, as well as the potential advantages to motivate its introduction and use in different settings.

We thank the reviewer and have revised the introduction.

Here is the new introduction, where particularly the first and last paragraphs have been rewritten:

"Genetic mapping studies of complex human traits have identified thousands of genetic loci implicated in the susceptibility to complex diseases [1, 2]. In other words, genome-wide association studies (GWAS) have linked thousands of single-nucleotide polymorphisms with complex human traits [1, 2]. While these findings have been beneficial for certain aspects, such as drug repositioning [1, 3-5], the identified individual genetic associations seldom exhibit large disease risks and explain a small portion of the calculated heritability for each trait [1, 6-8]. This implies that the genetic associations have a very poor predictive value [Hunter NEJM 2019]. Comprehensive interaction analyses between and among risk factors are an important tool to understand the genetic background of complex traits [7-11]. In general, interaction is declared when the combined effect’s magnitude of two or more factors is significantly different from the combined effect’s magnitude predicted by the model being tested [12- 14]. Nevertheless, it is challenging to address whether, and how genetic risk factors interact shaping human traits and to biologically interpret those interactions [9, 14]. Additionally, there is often confusion interpreting the results from different interaction models (i.e., additive interaction model and multiplicative interaction model). Therefore, in this study, we provide a framework of simulated scenarios that represent common and simple processes where two or more genetic factors may interplay in disease causation. We answer whether and how two or multiple genetic factors would interact in each simulated scenario. We also address those relationships in real data from a case-control study in rheumatoid arthritis (the EIRA study [15, 16]).

The association between individual genetic variants and an outcome (e.g., disease) is typically quantified as odds ratios (OR) or relative risks (RR). Often the case-control design is used to query low prevalence diseases, in which odds ratios approximate the relative risk in the population even though the samples are unevenly drawn. Interactions often examine two risk factors at a time, yielding three OR (or RR) notated as [13, 14]: OR11 for carrying both risk factors; OR10 and OR01 for the exclusive combinations, and OR00 for the absence of both risk factors, which is used as reference (OR00=1). Two different models are commonly used to test interaction, the additive (whose null hypothesis is OR11 = OR10 + OR01 – 1) and the multiplicative (whose null hypothesis is OR11 = OR10 � OR01). The additive model builds on the sufficient cause concept by KJ Rothman [12], who showed that if two factors are part of the disease’s cause and are part of the same sufficient cause (e.g., pathway), then their joint risks will be larger than their sum (often termed “departure from additivity”). This additive model has been criticized for always giving positive results [17, 18]. On the other hand, the multiplicative model has been criticized as a statistical convenience without a theoretical basis, boosted by the implicit multiplicativity in logistic regression [17, 19, 20]. There is often confusion regarding when and how to use and interpret each of the mentioned models. 

 In addition to evaluate the additive and multiplicative interaction models, we propose and study a third model, the multifactorial threshold model [21]. This model assumes that there is a minimum number of factors required for disease causation, which has a theoretical foundation from the concept of genetic liability [36]. Until now, no a statistical metric to account for the multifactorial threshold model is available, therefore we propose AMLES (additive-to-multiplicative linear effect scale), which compares number of risk factors to a threshold value. Although, the multifactorial threshold has been criticized for simplifying disease biology [21], our proposed metric could help to determine how well this model conforms to the biology.

Simulations are able to give an answer with high precision level if enough data points are generated. We design a variety of different scenarios as basis for the simulations. We build scenarios where we give the cause of the outcome (disease or otherwise) in its description. Our intension was to include the most likely real scenarios, and we provide practical examples from the literature. From the literature also comes the multifactorial threshold model, which is lacking a way to test for it in data. We also build scenarios where we model confounder situations, where there is no cause in the description yet there are apparent, false, risk factors. Again, these confounders are known in the literature. We will later show how the modelled scenarios can be distinguished in real data, and where they cannot. Across all simulations the risk factors are dichotomous, and neither necessary nor sufficient for disease to occur. While most of our simulated scenarios have two risk factors, we include modelling of more two factors and illustrate how it is possible to elucidate multifactorial interactions from a set of pairwise interactions. We compare to our simulation scenarios to additive and multiplicative risk scales, aiming to contribute to understand and interpret different models of interaction. This includes the multifactorial threshold model, and we find a metric that can measure fit not only to additive and multiplicative models, but also be used to elucidate the threshold level within the multifactorial threshold model from data."

Minor comments.

line 7-8. For synergy the two risk factors are required, rather than contribute.

We thank the reviewer for pointing this out and have removed that sentence from the abstract. As a result we now believe the message now is more clear.

Lines 25-26. Since until here you have been talking about multiple risk factors, it is good to mention that you are now referring to only 2 factors.

The sentence has been rephrased accordingly, now it reads like this: 

Interactions are often examined two risk factors at a time, yielding three odds (or risk) ratios notated as: OR11 for carrying both risk factors; OR10 and OR01 for the exclusive combinations.

Page 3. Y is universally used to refer to the outcome, I suggest using Z-X for referring to the 2 simultaneous exposures, and then another letter for the third exposure later.

The suggested change has been implemented in the text and the figures of the manuscript. The exposures are now named X, Z and V, respectively. 

Line 99. Principal components summarizing what?

The principal component analysis in GWAS is used to estimate the population structure and sample ancestry using the genotyping data. Since population structure can induce confounding factors in the genetic association studies, and in our study the interaction tests, the principal components are included as covariates. 

To clarify this, we have now re-structured the sentence the reviewer referred to as follows: 

In order to control by population stratification and differences between allele frequencies due to sex, the first ten principal components (which summarized the genotyping data) and sex were used as covariates in this analysis, respectively.

Line 433. I couldn’t find detailed figure legends. I think these should be useful, to present what the different panels are showing and provide a brief understanding of the figures without the need of reading the full manuscript

We apologize for this confusion. Rather than placing the detailed figure legends for the supplemental figures under "Supporting information captions" we had placed them at the bottom of the PDF files for the supplemental figures. We now have copied of the supplemental figure legends to the end of the manuscript. And corrected the naming of the supplemental files as instructed by the editor.

The main figures' legends are found embedded within the Results section, as per PLOS' author instructions, as captioned blocks.

---

## [Decision Letter · Decision Letter 1]

25 Jan 2021

PONE-D-20-24738R1

Understanding interactions between risk factors, and assessing the utility of the additive and multiplicative models through simulations

PLOS ONE

Dear Dr. Ramsköld,

Thank you for submitting your manuscript to PLOS ONE. After careful consideration, we feel that it has merit but does not fully meet PLOS ONE’s publication criteria as it currently stands. Therefore, we invite you to submit a revised version of the manuscript that addresses the points raised during the review process.

A **rebuttal letter** that responds to each point raised by the academic editor and reviewer(s). You should upload this letter as a separate file labeled 'Response to Reviewers'.A **marked-up copy** of your manuscript that highlights changes made to the original version. You should upload this as a separate file labeled 'Revised Manuscript with Track Changes'.An **unmarked version** of your revised paper without tracked changes. You should upload this as a separate file labeled 'Manuscript'.

We look forward to receiving your revised manuscript.

Kind regards,

Brecht Devleesschauwer

Academic Editor

PLOS ONE

Additional Editor Comments (if provided):

As indicated by the reviewer, we suggest you thoroughly copyedit your manuscript for language usage, spelling, and grammar. If you do not know anyone who can help you do this, you may wish to consider employing a professional scientific editing service.

- A copy of your manuscript showing your changes by either highlighting them or using track changes

- A clean copy of the edited manuscript

Reviewers' comments:

Reviewer's Responses to Questions

**Comments to the Author**

1. If the authors have adequately addressed your comments raised in a previous round of review and you feel that this manuscript is now acceptable for publication, you may indicate that here to bypass the “Comments to the Author” section, enter your conflict of interest statement in the “Confidential to Editor” section, and submit your "Accept" recommendation.

Reviewer #1: (No Response)

2. Is the manuscript technically sound, and do the data support the conclusions?

Reviewer #1: Yes

3. Has the statistical analysis been performed appropriately and rigorously? 

Reviewer #1: Yes

4. Have the authors made all data underlying the findings in their manuscript fully available?

Reviewer #1: Yes

5. Is the manuscript presented in an intelligible fashion and written in standard English?

Reviewer #1: No

6. Review Comments to the Author

Reviewer #1: The paper has greatly improved. I only strongly recommend a thorough language/grammar revision as many sentences added to the revised version can be improved

7. PLOS authors have the option to publish the peer review history of their article (what does this mean?). If published, this will include your full peer review and any attached files.

Reviewer #1: No

---

## [Author Response · Author response to Decision Letter 1]

10 Mar 2021

The manuscript has gone through copy editing since last submission, by the authors and by a professional editing service (Janet Ahlberg).

---

## [Editor Report · Decision Letter 2]

5 Apr 2021

Understanding interactions between risk factors, and assessing the utility of the additive and multiplicative models through simulations

PONE-D-20-24738R2

Dear Dr. Ramsköld,

We’re pleased to inform you that your manuscript has been judged scientifically suitable for publication and will be formally accepted for publication once it meets all outstanding technical requirements.

Kind regards,

Brecht Devleesschauwer

Academic Editor

PLOS ONE
---

## [Editor Report · Acceptance letter]

15 Apr 2021

PONE-D-20-24738R2 

Understanding interactions between risk factors, and assessing the utility of the additive and multiplicative models through simulations 

Dear Dr. Ramsköld:

I'm pleased to inform you that your manuscript has been deemed suitable for publication in PLOS ONE. Congratulations! Your manuscript is now with our production department. 

Kind regards, 

on behalf of

Prof. Dr. Brecht Devleesschauwer 

Academic Editor

PLOS ONE